# Rapid and High Throughput Hydroponics Phenotyping Method for Evaluating Chickpea Resistance to Phytophthora Root Rot

**DOI:** 10.3390/plants12234069

**Published:** 2023-12-04

**Authors:** Muhammad A. Asif, Sean L. Bithell, Ramethaa Pirathiban, Brian R. Cullis, David Glyn Dionaldo Hughes, Aidan McGarty, Nicole Dron, Kristy Hobson

**Affiliations:** 1Chickpea Breeding Australia, New South Wales Department of Primary Industries, Tamworth Agricultural Institute, Tamworth, NSW 2340, Australia; kristy.hobson@dpi.nsw.gov.au; 2New South Wales Department of Primary Industries, Tamworth Agricultural Institute, Tamworth, NSW 2340, Australia; sean.bithell@dpi.nsw.gov.au (S.L.B.); nicole.dron@dpi.nsw.gov.au (N.D.); 3Mixed Models and Experimental Design Lab, National Institute for Applied Statistics and Research Australia, School of Mathematics and Applied Statistics, University of Wollongong, Wollongong, NSW 2522, Australiahughesd@uow.edu.au (D.G.D.H.);

**Keywords:** Phytophthora root rot, chickpea, hydroponics screening method, high throughput, plant breeding, root disease resistance breeding, quantitative resistance phenotyping

## Abstract

Phytophthora root rot (PRR) is a major constraint to chickpea production in Australia. Management options for controlling the disease are limited to crop rotation and avoiding high risk paddocks for planting. Current Australian cultivars have partial PRR resistance, and new sources of resistance are needed to breed cultivars with improved resistance. Field- and glasshouse-based PRR resistance phenotyping methods are labour intensive, time consuming, and provide seasonally variable results; hence, these methods limit breeding programs’ abilities to screen large numbers of genotypes. In this study, we developed a new space saving (400 plants/m^2^), rapid (<12 days), and simplified hydroponics-based PRR phenotyping method, which eliminated seedling transplant requirements following germination and preparation of zoospore inoculum. The method also provided post-phenotyping propagation all the way through to seed production for selected high-resistance lines. A test of 11 diverse chickpea genotypes provided both qualitative (PRR symptoms) and quantitative (amount of pathogen DNA in roots) results demonstrating that the method successfully differentiated between genotypes with differing PRR resistance. Furthermore, PRR resistance hydroponic assessment results for 180 recombinant inbred lines (RILs) were correlated strongly with the field-based phenotyping, indicating the field phenotype relevance of this method. Finally, post-phenotyping high-resistance genotypes were selected. These were successfully transplanted and propagated all the way through to seed production; this demonstrated the utility of the rapid hydroponics method (RHM) for selection of individuals from segregating populations. The RHM will facilitate the rapid identification and propagation of new PRR resistance sources, especially in large breeding populations at early evaluation stages.

## 1. Introduction

Phytophthora root rot (PRR) caused by an oomycete pathogen *Phytophthora medicaginis* is one of the major soil-borne diseases of chickpea in Australia. In 2012, PRR was estimated to cost chickpea growers up to $8.2 million per year; however, based on current production costs, economic losses are thought to be higher [1,2]. The disease was first reported in Australia in the 1980s and is now commonly found in major chickpea growing regions, including northern New South Wales and southern Queensland [2,3]. The typical above-ground symptoms of infected plants include wilting and chlorosis, which can lead to defoliation. The dark brown lesions on roots lead to decay of lateral and tap roots; lesions can extend up the epicotyl above ground level to form a stem constriction or canker [3,4]. High soil moisture followed by rainfall provides a conducive environment for PRR disease development; hence, the yield losses are higher during wet and waterlogged conditions than in normal growing seasons [5,6]. In the northern Australian chickpea growing region, the cultivar Yorker can lose 30–50% yield from PRR under dryland conditions and up to 70% loss in high rainfall seasons [5]. Hence, crop losses to PRR are a major constraint to expansion of the chickpea growing area and productivity in northern Australia. 

Management of PRR is a challenging task due to the long survival (up to 10 years) of the resting pathogen structure (oospores) in the soil and the ability of the pathogen to cause infection in plants at any growth stage [7,8]. Currently, growers have limited options for disease control, which include planting partially resistant cultivars, crop rotation, and avoiding high-risk paddocks. Chemical control using a metalaxyl-based seed treatment only provides control for a limited time (6–8 weeks) and is not cost effective. Biocontrol approaches such as antagonistic root-associated bacteria [9], endophytic actinobacteria [10], and rhizobia *Mesorhizobium ciceri* [11] have shown some potential to contribute to management of PRR in chickpea. However, this research is in the early stage of development and requires field validation of glasshouse results. Of the different PRR control measures, breeding disease-resistant cultivars is one of the most effective and economical methods.

Through extensive breeding efforts, the level of resistance of current Australian chickpea cultivars such as Kyabra, Jimbour, Moti, and Yorker have improved through incorporating the resistance identified in chickpea accession (CPI 56564 = ICC11870) [12,13]. Likewise, the resistance from a wild relative of chickpea (*Cicer echinospermum*) has been explored and incorporated into cultivated *C. arietinum* background to generate interspecific hybrids [4]. A PRR-resistant derivative of wild chickpea (*C. echinospermum*, interspecific breeding line 04067-81-2-1-1) was used to develop two interspecific recombinant inbred line (RIL) mapping populations (Rupali × 04067-81-2-1-1 and Yorker × 04067-81-2-1-1). Quantitative trait locus (QTL) analysis of these mapping populations resulted in four major QTL for PRR resistance on chromosomes 3 and 6 [14]. The progress in developing new PRR-resistant cultivars is slow due to limited availability of PRR resistance within the cultivated chickpea species (*C. arietinum*) and challenges, like linkage drag of undesirable agronomical traits associated with use of available resistance from *C. echinospermum* [15,16]. Therefore, more research is needed to identify new PRR-resistant sources to develop resistant cultivars.

Previous selection methods for identifying PRR resistance in chickpea relied primarily on field evaluations or on glasshouse seedling-based tests [4,5,17,18]. Although field screening is conducted under natural conditions, the process is labour intensive, time consuming, and heavily affected by seasonal conditions. The glasshouse soil-based screening methods are low throughput and showed inconsistent results and discrepancies between glasshouse and field PRR-resistance rankings of genotypes [4]. Hence, these methods are not ideal for screening a large number of genotypes at the early stages of a breeding program. 

Recent work in chickpea [19,20] and soybean [21] showed that a hydroponics screening is reliable and a rapid PRR phenotyping method. There are several potential advantages to hydroponic-based resistance screening over soil screenings. Hydroponics methods can ensure plants are exposed to uniform disease pressure and can be grown in high densities, providing the ability for high-throughput resistance screenings [19,22,23]. Due to the relatively low seed number requirements of hydroponic methods (1–3 seeds) in comparison to field disease assessment screenings (20 seeds/plot), hydroponics can provide the ability to screen lines at an earlier stage. This early-stage screening may reduce the number of lines to carry over to the next breeding stage by excluding lines with poor PRR phenotypes and so improve breeding efficiency.

A number of previous PRR hydroponics methods have some limitations for screening larger populations in breeding programs, including (1) the use of small tanks (10, 12.5, 60 L) [19,20,21] or pots (2.1, 4.5 L) [19,22] to hold the growth solution; (2) reduced scalability in terms of the space required to screen large populations [20,22]; (3) reliance on a two-step process (seed germination followed by transplanting into tanks/pots), which lengthens and complicates the procedure [19,21,22]; (5) the use of zoospores over mycelial-oospore *P. medicaginis* inoculum, which introduces additional preparations [19]; and, (6) in addition, we could not find reports for methods that evaluated the ability to utilise plant selections from the phenotyping process for post-phenotyping propagation. The ability to propagate elite-performing lines may be important as both a time-saving method and ensuring that the correct genotypes are being selected. Whilst these methods met the needs of small-scale experiments, we sought to develop a simplified, space saving, reliable, and rapid PRR-resistance phenotyping method for screening large numbers of genotypes, hereafter referred to as the rapid hydroponic method (RHM). 

Through modifications to previous protocols, we simplified the process by directly germinating seed and growing plants on the same racks without transplanting, achieved high throughput by propagating a high density of plants (400 plants/m^2^), and provided rapid PRR-resistance phenotyping with final results 12 days after inoculation. The new method was tested first through an experiment with 11 chickpea genotypes that differ for PRR disease-resistance ratings (Experiment 1—E1). Furthermore, a RIL mapping population (180 RILs) was screened to determine the correlation between PRR parameters derived from the hydroponics (Experiment 2—E2) and previously published traits obtained from a field experiment [14]. Finally, the hydroponic test-grown plants were transplanted into soil to test the ability of this method to select and propagate individuals from segregating populations. We demonstrated that the current method is reliable, is easy to set up, and can greatly increase the efficiency of PRR-resistance screening in chickpea. 

## 2. Results

### 2.1. Hydroponics Phenotyping: E1

A set of 11 diverse chickpea genotypes were tested for their response to PRR resistance in a newly developed hydroponics system. Visible disease symptoms showed clear dissimilarity between the most susceptible and most resistant genotypes, Rupali and 04067-81-2-1-1, respectively. Foliage wilt and root discolouration were apparent on Rupali six days post inoculation, and these symptoms progressed to high levels of leaf chlorosis and brown-to-black root lesions 12 days post inoculation (Figure 1). In contrast, 04067-81-2-1-1 showed minimal foliar symptoms, no epicotyl canker, and minor root infection over the duration of the experiment (Figure 1). Overall, the susceptible genotypes displayed a wide range of symptoms such as leaf chlorosis, stem cankers, root discolouration, reduced root system development, and plant death. By contrast, the moderately resistant to moderately susceptible (MR-MS) genotype 04067-81-2-1-1 and other *C. echinospermum*-crossed genotypes CICA1328 (pedigree: PBA HatTrick/04067-89) and CICA1815 (04067-81-2-1-1/CICA0913//CICA1003) had minimum leaf chlorosis, and their roots were less diseased than those of susceptible genotypes.

The distribution of PRR disease symptoms across the tank was highly uniform, and no significant spatial variation was observed. Table 1 presents a summary of the sources of variation for each random term in the generalised linear mixed model (GLMM) for each of the four parameters. These values were derived from the Penalized Quasi Likelihood (PQL) estimates of the variance components and are expressed as a part of the total variance, excluding the baseline variance of π^2^β. The dominant source of variation was attributed to genotype (Table 1). Some additional non-genetic variations were present for most parameters, but these were not consistent or substantial, reflecting the relatively homogeneous disease incidence across the experiment.

The predicted genotypic effects (on the logit scale) and back-transformed probabilities for each of the three parameters aligned well to their field-based disease-resistance ratings (Table 2). The lower predicted and back-transformed values for leaf chlorosis incidence and canker incidence indicated a lower disease level (leaf chlorosis and canker development). Results showed that the MR-MS genotype 04067-81-2-1-1 had the lowest predicted leaf chlorosis incidence probability of 0.26. In contrast, the very susceptible (VS) Rupali had a higher leaf-chlorosis-incidence predicted probability of 0.79. Overall, the susceptible (S) and VS genotypes leaf-chlorosis-incidence predicted probabilities were between 0.62–0.79 compared to 0.42–0.57 for moderately susceptible (MS) genotypes (Table 2). Likewise, canker development on the epicotyl was also higher in susceptible genotypes than in resistant genotypes (Table 2). The most susceptible genotype, Rupali, showed a predicted probability of canker incidence of 0.81 compared to 0.07 for the most resistant genotype (04067-81-2-1-1) (Table 2). All nine other genotypes had predicted probabilities of canker incidence between 0.28–0.87. Surprisingly, CICA1328 had a predicted canker-incidence probability of 0.72 despite having lower (0.42) leaf-chlorosis-incidence predicted probability whereas Kyabra, the S-VS genotype, had the lowest (0.47) predicted canker-incidence probability among all the susceptible genotypes (Table 2). 

Like the canker development, all the genotypes had root lesions, but the symptom severity as represented by the root lesion score was more severe in susceptible genotypes. Unlike the leaf-chlorosis incidence and canker incidence, the higher predicted genotypic effects and associated 50% threshold probability values indicate lower disease level as these values correspond to the predicted probabilities for a score of 1 or 2 from a 1–4 score range. It is evident from the results that the most resistant genotype, 04067-81-2-1-1, had the highest predicted probability (0.95) of having a disease score below 2 followed by CICA1815 (0.32) and CICA1328 (0.42) with all other susceptible genotypes ranging from 0–0.08 (Table 2). 

Spearman rank correlations of predicted genotypic effects provided strong correlations among all measured parameters. In particular, leaf-chlorosis incidence was highly correlated with the root-lesion score (−0.97) and canker incidence (*r* = 0.78), while root-lesion severity (−0.82) was also strongly correlated with canker incidence (Table 3). 

### 2.2. Molecular Quantification of the Pathogen in Root Samples

The roots of three genotypes (04067-81-2-1-1, Yorker, Rupali) harvested at the end of E1 had *P. medicaginis* DNA concentrations quantified. The MR-MS 04067-81-2-1-1 had significantly lower *P. medicaginis* DNA concentrations per gram of root than the other two genotypes (Figure 2). The amount of DNA in Yorker, although slightly lower than the most susceptible genotype Rupali, did not differ significantly (Figure 2). 

### 2.3. Post-Phenotyping Propagation

After the cessation of E1, plants of genotype 04067-81-2-1-1 were transplanted into potting mix to test post-phenotyping propagation (Appendix A). The 04067-81-2-1-1 is the most PRR-resistant *C. echinospermum*–chickpea backcross genotype available in the CBA gene pool and is a benchmark for future selection. Hence, this genotype was selected for optimising the transplanting process for future single plant selections. Transplanted plants initially showed signs of stress and dropped some older leaves, but after one week, they initiated new leaf production, and after 6–8 weeks they produced flowers and pods (Appendix A).

### 2.4. Hydroponics Phenotyping: E2

A RIL mapping population (180 RILs) and both parents were screened in a hydroponics system with the aim to determine a relationship between PRR parameters derived from the hydroponics and traits obtained from field experiment. Eleven days after inoculation with *P. medicaginis*, the MR-MS RIL parent (04067-81-2-1-1) was green and healthy, whereas the susceptible parent (Yorker) showed severe PRR disease symptoms (leaf chlorosis and root discolouration) (Appendix A). Genotype 04067-81-2-1-1 had a lower predicted value for leaf chlorosis incidence (0.22 ± 0.03 SE) and was more resistant than Yorker (0.65 ± 0.04 SE). The results for the RIL population demonstrated a continuous phenotype distribution, with leaf chlorosis incidence values ranging from 0.22 to 0.65 (Figure 3a).

### 2.5. Field Phenotyping 

The field disease data of the RIL population included the number of dead and chlorotic plants at three time points. These were combined to calculate the expected lifetime for each RIL. Plants with longer expected lifetimes were more PRR-resistant as generally fewer occurrences of chlorosis and or death were observed. Field results for the MR-MS parent, 04067-81-2-1-1, showed a larger predicted value for expected lifetime (9.4) compared to the more susceptible Yorker (2.1), which died early due to severe PRR disease (Figure 3b). Predicted expected lifetime values ranged from 2.1–9.4.

### 2.6. Correlation between Hydroponic (E2) and Field Experiments

A primary aim of this study was to compare phenotypic data based on leaf chlorosis and death (expected lifetime) from the field experiment [14] and leaf chlorosis incidence from a hydroponics experiment (E2). This was achieved through the use of a binomial-normal (bivariate) GLMM, to model both sets of phenotypic data and capture any genetic correlation present. Genetic effects by parameter comprised both additive and non-additive effects, whereby an unstructured (US) variance model was fit for each. Results from PQL estimation showed a strong negative genetic correlation between hydroponics (E2) and field data (Table 4), with an additive genetic correlation estimated at −1 and a total genetic correlation estimated at −0.48. Negative correlations are expected, since, for the field experiment, higher expected lifetime values indicate higher disease resistance, whereas for the hydroponic experiment lower leaf-chlorosis-incidence values indicate higher disease resistance.

## 3. Discussion 

The primary aim of this study was to develop a reliable high-throughput method for PRR-resistance screening in chickpea. Results showed the new method can efficiently differentiate between PRR-resistant and susceptible genotypes. Post-germination transplanting, required in previous PRR hydroponics protocols, has been circumvented by directly germinating and growing plants on racks. Furthermore, the preparation of zoospores has been avoided though the use of mycelial-oospore inoculum that resulted in rapid infection and symptom development. This makes the new method simpler and quicker as the significant amount of time and labour required for transplanting and propagating zoospores is mitigated. Final PRR disease assessments of chickpea seedlings were provided 11–12 days after inoculation, in contrast to the 20–24 post-inoculation days required for hydroponic-based survival time analyses reported by Amalraj et al. [19]. In other small scale hydroponics studies on alfalfa and soybean root disease, symptoms were assessed in 14 and 21 days, respectively [21,22]. The plant density of our method (330 plants/tank, 400 plants/m^2^) is higher than that reported for any prior hydroponic-based PRR phenotyping studies [19,21]. Finally, the strong correlation between PRR-resistance parameters derived from the hydroponic- and field-based phenotyping demonstrates that the method provides field-relevant results.

PRR disease severity is often visually scored by an ordinal (1–9) rating scale in chickpea [3,4,6,18,26]. In field phenotyping, the disease incidence is usually based on the number of diseased to total plants per plot [4,5,18]. The assessment of single plants in pots and/or hydroponics experiments requires different methods. Here, we used a logit-based incidence analysis based on the number of chlorotic to total leaves per plant for disease assessments. This assessment method provided good separation among genotypes of differing PRR resistance and could be converted to a 1–9 score for ease of comparing it with published scores as they are only available on an ordinal scale. The RHM provides the ability to select phenotypically desirable (i.e., the most PRR-resistant) lines, as shown for 04067-81-2-1-1 in our first experiment. The selection process is rapid as most of the susceptible and moderately susceptible lines have distinct leaf and root disease symptoms by the end of the experiment. 

For key cultivars with a known range of PRR field-resistance reactions, the RHM provided field-relevant results. The RHM disease scores (leaf and root parameters) aligned well with published cultivar ratings and clearly discriminated between the different varieties of PRR-resistance rating groups (MR-MS, S, and VS). Results showed that 04067-81-2-1-1 and CICA1328 exhibited an improved level of PRR resistance in comparison to Yorker and Rupali; this was also supported by previous findings for glasshouse [20,27] and field screenings [14]. Differential genotype responses were further confirmed through quantification of *P. medicaginis* DNA concentrations in the roots, which showed significantly lower pathogen DNA concentrations in the MR-MS genotype (04067-81-2-1-1) than in the more susceptible genotypes, Rupali and Yorker. The *P. medicaginis* DNA quantification in each genotype was in accordance with observations of PRR parameters measured in hydroponics E1 and with known levels of PRR resistance [28]. These results provided quantitative evidence of the phenotypic responses of the host plant genotypes to PRR infection in our phenotyping method. 

Strong correlations were observed for all measured parameters in E1. Previous studies in chickpea also found a similar relationship among different resistance parameters. In a recent hydroponics study, plant survival (Kaplan–Meier estimates survival) and canker length were highly correlated in intraspecific (Yorker × Genesis 114) and interspecific (04067-81-2-1-1 × Rupali) RIL mapping populations [19]. Significant correlation between chlorosis incidence parameters and other PRR-related parameters in current and previous studies suggests that different shoot- and root-related resistance parameters could be under similar genetic control. Evidence for linked parameters was further provided by the presence of a co-located QTL for plant survival and canker length on chromosomes 4 and 7 in a 04067-81-2-1-1 × Rupali RIL population [19]. 

RILs and the parent (04067-81-2-1-1) with low disease scores in the hydroponics experiment (higher resistance level) also showed high expected lifetimes in the field, which hence was indicative of the similar performance of RILs under hydroponics and field conditions. Successful validation of field-based screening results in current and previous studies in chickpea [19] and alfalfa [22] proved that hydroponic-based phenotyping methods can offer accurate and reproducible phenotyping for studying root disease resistance. A key feature of quantitative resistance disease reactions in a genetically segregating population is a continuous (non-categorical) distribution of disease phenotypes [29]. We demonstrated that the RIL population showed a continuous distribution for chlorosis reactions and that the RHM could provide a range of disease reactions among the heterogeneous RIL.

Our RHM was successful for PRR phenotyping; however, a number of factors can influence a plant’s response to PRR in hydroponics. Plants growing in hydroponics require a continuous oxygen supply into the nutrient media for optimum root growth [30,31]. Anaerobic root conditions can result in similar foliar symptoms as PRR including wilting and leaf chlorosis [32]. Chickpea genotypes (Rupali, Yorker, and 04067-81-2-1-1) growing under the combination of hypoxia and *P. medicaginis* infection showed more foliar symptoms and root disease than aerated infected plants [6]. Likewise, tomato plants were more susceptible to Pythium root rot (*Pythium* spp.) and showed significantly reduced shoot and root growth while growing under low oxygen supply (0.4–0.7 ppm) [31]. To mitigate the hypoxia effect, we continuously aerated the nutrient media using an air pump and also used susceptible and resistant checks, which provided the ability to identify the effects of other factors separate from PRR disease effects. 

The RHM described in this study is suitable for screening a large number of chickpea genotypes for PRR resistance in a limited space over a short period of time. Another key feature of our method is the propagation ability of selected material. We demonstrated the successful flowering and seed set of selected genotypes following transplanting. This feature allows for the recovery of elite-performing individuals from early generation segregating populations (e.g., F_2_ or F_3_) to progress to seed production whilst also freeing up the hydroponic tanks for subsequent screenings. Finally, the new method represents an efficient and reliable system for the evaluation of PRR resistance in chickpea germplasm and has the potential to be used in other crops to study root diseases like Fusarium root rot, Pythium root rot, and Rhizoctonia root rot [23].

## 4. Materials and Methods

### 4.1. Plant Material

Two hydroponics experiments (E1 and E2) were conducted using the RHM as described later. In E1, a set of 11 chickpea genotypes, having a wide range of PRR-resistance responses, were selected to assess the reliability and accuracy of the RHM for PRR screening. Resistance ratings were accessed from the Grains Research and Development Corporation, National Variety Trial (NVT) rating program [25]. The NVT program adopted a new rating system in 2020 based on new resistance rating definitions for pulses. Current NVT ratings were not available for all genotypes; the selection of 6 of the 11 genotypes with no current NVT resistance ratings was made using published [5] and unpublished PRR yield response data (Hobson and Bithell, unpublished results).

E2 tested 180 F_6_-derived mapping population RILs (Yorker × 04067-81-2-1-1), which had been phenotyped in a field experiment for PRR resistance [14] along with two parents and a PRR-susceptible control (Rupali). The RIL population was developed by crossing an Australian PRR-susceptible variety, Yorker (pedigree: 8507-28H/946-31), with a MR-MS interspecific breeding line, 04067-81-2-1-1 (a backcross derivative from *C. echinospermum*: Howzat/ILWC 245//99039-1013), from the National Chickpea Breeding program (New South Wales Department of Primary Industries, Tamworth, Australia). 

### 4.2. Hydroponics Phenotyping (E1 and E2)

#### 4.2.1. The Phenotyping Setup

The hydroponics setup consisted of a tank and seed racks (Appendix A). The polypropylene tank (1200L × 900W × 600H mm) was fitted with an air pump (Hailea, Model ACO-328) and connected with six round ceramic air stones (25 mm diameter) to provide adequate aeration to nutrient media (Appendix A). The air pump was controlled by a 24 h timer, set to a 45/15 min on/off cycle for each hour. Seed racks were prepared by using the Wearex™ UHMWPE (Ultra-High Molecular Weight Polyethylene, Allplastics Engineering Pty Ltd., Chatswood, Australia) sheet (400L × 350W × 12H mm). Each rack had 56 (25 mm) diameter holes in an 8-row by 7-column rectangular array. Racks were fitted with a 6 mm plastic gutter guard mesh (Whites StayMesh™ Gutter Kit, Whites Group Pty Ltd., Pemulwuy, Australia) to hold seeds (Appendix A). Six racks were placed in a tank in 2 × 3 rectangular arrays. Of the 56 holes in each rack, only 55 holes were available for plants as the middle hole in each rack was kept empty to monitor media level and pH measurement. Each rack had four detachable stainless-steel legs (190 mm) that were attached to the racks when required for plant growth (Appendix A). The tank, seed racks, and stainless-steel legs were washed with commercial bleach (0.042% (*w*/*v*) sodium hypochlorite (NaOCl) and sprayed with 70% ethanol prior to their use, to eliminate contamination. The experiments were undertaken in a growth room (Percival Scientific AR-119 LED4) located at the Tamworth Agricultural Institute (lat: 31°08′55.3″ S; long: 150°58′53.1″ E) in Tamworth, New South Wales. A single tank with six racks was used for E1, whereas two tanks situated side by side in a growth room were used for the E2 using the method described below. 

#### 4.2.2. Experimental Procedure 

Three uniform sized seeds of each genotype were sterilised in 1% NaOCl (*w*/*v*) for 15 min and rinsed with reverse osmosis (RO) water three times. Seeds were placed in each hole of the rack according to their positions in the experimental design. Seeds were imbibed overnight in an aerated solution of 0.5 mM CaSO_4_ in darkness (Appendix A) and maintained under controlled conditions (25/15 ± 2 °C day/night, 50/70% day/night relative humidity) in a growth room. After 24 h, the CaSO_4_ was replaced with 10% concentration aerated nutrient solution. The nutrient solution contained (in mM): 5.0 Ca^2+^, 3.75 K^+^, 3.125 NH^4+^, 0.4 Mg^2+^, 0.2 Na^+^, 5.4 SO_4_^2−^, 6.875 NO_3_^−^, 0.2 H_2_PO_4_^−^, 0.1 SiO_3_^2−^, 0.1 Fe-sequestrene, 0.05 Cl^−^, 0.025 BO_3_^3−^, 0.002 Mn^2+^, 0.002 Zn^2+^, 0.0005 Cu^2+^, 0.0005 MoO_4_^2−^, 0.001 Ni^2+^, and 1.0 MES (2-[*N*-Morpholino] ethane sulfonic acid) [19,33,34]. KOH was used to adjust the pH to 6.5. pH was measured daily and maintained at the desired level throughout the duration of the experiment. Seeds were kept on the media for three days in the dark, and seedlings were then transferred to the 25% concentration aerated nutrient solution and exposed to light (13/11 h light/dark). Seedlings were thinned to one seedling per hole at the one-node leaf stage and held in position with a 30 × 30 mm sponge strip (~5 mm thick) (Appendix A). The sponge strips were sterilised by soaking them in 0.042% (*w*/*v*) sodium hypochlorite for 2 h followed by rinsing with RO water and spraying with 70% ethanol, prior to use. When plants were at the two-node stage, legs were attached to the racks, and the tub was filled with 175 L 25% concentration aerated nutrient solution. The gaps between the racks and tank walls were covered with black PVC sheet cuttings (Suntuf PVC Handisheet, Ourgreen, Nanjing, China) to maintain a dark environment in the tank to promote root growth and prevent growth of algae (Appendix A). At the three-node stage, plants were inoculated with *P. medicaginis* mycelial oospore suspension to provide a concentration of 13 oospores/mL in each tank. The 13 oospores/mL concentration was used, as pilot studies comparing 13 and 25 oospores/mL inoculum concentrations showed no significant difference for leaf-chlorosis scores between these two concentrations (results not presented). The day of inoculation counted as day 1, and the day before rating was counted as the final day of screening. Plants were examined daily after inoculation for PRR symptoms including wilting, leaf chlorosis, and canker development and were rated when the susceptible genotype (Rupali) showed 70–80% leaf chlorosis (Appendix A). The general workflow for the hydroponics system is summarised in Appendix A.

### 4.3. Inoculum Production

An aggressive *P. medicaginis* isolate (7842) was selected from a set of 37 isolates for use in this study [27]. Inoculum was produced as described by Dron et al. [20]. The isolate was cultured on V8 agar with 2.5% CaCO_3_ and was grown for six weeks at 25 °C in the dark. A mycelium oospore suspension was prepared from the plates by scraping agar culture plates with matured oospores into a beaker. Sterile water of approximately 10% by volume of the agar was added and blended using hand-held Braun 600 W blenders for approximately three min, until homogeneous. Average oospore concentrations were determined using counts under a 20 × 50 mm coverslip using a microscope. The total amount of inoculum required was split into six equal portions and was applied to each of the six racks through an empty hole to achieve uniform inoculum distribution.

### 4.4. Measurement of PRR-Related Parameters

All plants were individually assessed at the end of the experiment (12 days after inoculation for E1 and 11 days after inoculation for E2). Three parameters were measured for E1, whereas only the first two parameters were measured for E2, as only the above ground parameters were measured for RILs in the field study [14].

Leaf-chlorosis incidence: The total number of leaves and number of chlorotic leaves were counted on each plant. Leaves were categorised as chlorotic if 50% or more of leaflets on a leaf were chlorotic.Canker incidence: For plants that developed epicotyl cankers, the canker length starting from the epicotyl region and proceeding upwards on the stem was measured using an electronic digital 0–150 mm Vernier calliper. The canker length was then used to calculate the canker incidence.Root lesion score: Dark brown to black root lesions were scored on a 1–4 scale, where 1 = 0–25% root browning/death and 4 = 75–100% browning/death as described by Pratt et al. [35].

### 4.5. Molecular Quantification of the Pathogen in Root Samples

Roots of three genotypes, 04067-81-2-1-1, Yorker, and Rupali, representing three distinct groups of resistance (MR-MS, S, and VS, respectively), were harvested at the end of E1, dried at 40 °C for 72 h, weighed, placed in 200 g of sand, and sent to the South Australian Research and Development Institute (Adelaide, Australia) to quantify *P. medicaginis* DNA concentration via qPCR as described by Bithell et al. [5]. The root weight values were used to normalise the DNA concentrations relative to the different sized root systems. 

### 4.6. Post-Phenotyping Propagation

Three plants of MR-MS genotype (04067-81-2-1-1) were randomly selected at the end of E1 to assess their post-phenotyping propagation ability. The roots were gently washed with RO water followed by treatment with Apron XL 350 ES, Syngenta Crop Protection PtY Limited, Macquarie Park, Australia (350 g/L Metalaxyl-M) for 1 min. Plants were then transplanted into a free draining pot (20 cm height × 20 cm diameter) filled with 2.5 kg of commercial potting mix and watered with 25% hydroponics nutrient solution for four days. They were kept in the growth room for four days to minimise transplanting shock, then shifted to a glasshouse and watered with RO water. Plants were harvested at physiological maturity after seed maturation. 

### 4.7. Field Phenotyping

The Yorker × 04067-81-2-1-1 RIL population was phenotyped in a field experiment at the Hermitage Research Facility, Warwick, Queensland, Australia (latitude: 28.21° S) in 2015 under irrigated conditions as described by Amalraj et al. [14]. RILs and parents were replicated four times. Twenty seeds of each line were sown in a 1.2 m long single row plot, and a *P. medicaginis* oospore mycelial suspension consisting of equal concentrations of oospores from ten isolates were applied in-furrow at planting. Following emergence counts, the number of chlorotic and dead plants were recorded at three time points and combined to form a new measure: “unhealthy plants” for each time interval. This was used in the subsequent formulation of expected lifetimes. An initial emergence count was conducted on 28 July 2015 (day 0), and subsequent assessments were conducted on 20 August 2015 (day 24), 13 October 2015 (day 78), and 4 November 2015 (day 100), with an average interval of 33 days between assessments.

### 4.8. Experimental Design

E1 contained a single tank with six racks placed in a 3 × 2 array with 55 usable holes in an 8 × 7 array in each rack. Five replicates of each genotype were allocated to plots (holes) within each rack, and in total 30 replicates/tank were used. The allocation of holes to genotypes was determined using the model-based design software ODW V1 [36]. The mixed model used to drive the design search included terms for racks, columns, and rows within racks and genotypes. All terms were included in the model as random effects. The resulting allocation produced a design in which racks and genotypes were orthogonal, that is, five holes per rack for each genotype.

Two adjacent tanks were used in E2. Each tank had an identical layout to the tank used in E1. Since there were 660 holes and 183 genotypes, utilization of all holes would result in an unequally replicated design, with 103 genotypes allocated to four holes and 80 genotypes allocated to three holes. Rather than randomly assigning genotypes to replication state, the genomic relationship matrix (GRM) using the marker data (2061 SNP markers) available through Amalraj et al. [14] was used to determine the level of replication for each genotype, which was either three or four times across the two tanks. The allocation of holes to genotypes was then achieved using a similar approach to E1, except additional non-genetic effects were included in the LMM to account for the two tanks. The total genetic effects were linked to the A-optimality criteria in the design search.

### 4.9. Statistical Analysis

For the multinomial GLMM, the cumulative probabilities rather than the all probabilities were modelled given the ordinal multinomial response [37]. The canker incidence parameter was derived from canker length being a binary response with value (0, 1), 0 being length = 0 and 1 otherwise. 

#### 4.9.1. Field Phenotyping

Using the total number of emerged plants and total number of unhealthy plants on each plot, a binomial generalised linear model (GLM) was formed using a complementary log-log link function as in Bartlett [38], using the formula:$$\ln [-\ln\left(1-p_{ij}\right)]=\beta_{i}+\gamma_{j}$$
where $\beta_{i}$ $\left(i=1,\ldots ,896\right)$ $\left(\beta_{1}=0\right)$ are the plot effects, and $\gamma_{j}$ $\left(j=1,2,3\right)$ are the time effects. Note that the conditional probabilities $p_{ij}$ are the probability that a plant on the ith plot will become unhealthy in the $\left(t_{j-1},t_{j}\right]$ interval given that it was healthy at time $t_{j-1}$. To calculate expected lifetimes, a reparameterization of the model involving the $\gamma_{j}$s was made, namely, $\gamma_{j}=\alpha +\ln\left(e^{\theta t_{j}}-e^{\theta t_{j-1}}\right)\;note$ $t_{0}=0$. This involved fixing θ and estimating the remaining parameters in the model. The resulting deviance of the model was assessed for a range of θ values to determine $\hat{\theta }$ corresponding to the minimum deviance. Finally, the expected lifetime for each plot was calculated using the formula:$$\hat{\mu }i=\frac{-exp [exp\left(\hat{\alpha }+\hat{\beta }i\right)]\left(C+\hat{\alpha }+\hat{\beta }i+\sum_{k=1}^{\infty }\frac{{ [-exp\left(\hat{\alpha }+\hat{\beta }i\right)]}^{k}}{kk!}\right)}{\hat{\theta }}$$
where C = 0.577216 is Euler’s constant [38,39].

#### 4.9.2. Combined Analysis of Hydroponics (E2) and Field Experiments

Two parameters, leaf-chlorosis incidence from E2 and expected lifetime from the field experiment, were analysed using a bivariate binomial–normal GLMM. The aim of this analysis was to assess the agreement between PRR-resistance assessment using the RHM and field-based approaches. The focal component of the (bivariate) GLMM was the choice of variance model for the set of additive and non-additive genetics by parameter effects. The proposed model for modelling the between-parameter genetic variance includes a variance for each parameter and a (non-zero) covariance. This model is referred to as the unstructured (US) variance model and this model was fitted to the additive variety by trait (field and hydroponics) effects and the non-additive variety by trait effects, respectively. As the proposed model utilised PQL, a parametric bootstrap-based approach was used to assess the model fit. This involved simulating data from the fitted model and assessing the sampling distribution of the between-parameter genetic variance components from the simulated data, most notably the total genetic correlation between parameters. The parametric bootstrap included 2000 simulated datasets; of these, 726 were omitted due to non-convergence. The remaining 1274 simulations resulted in an average total genetic correlation of −0.54 with an associated 95% confidence interval [−0.70,−0.36]. The GLMM also included a set of baseline terms, which accounted for the plot structures of each experiment. All analyses were undertaken using ASREML-R V4 [40].

## 5. Conclusions

The current study describes a rapid and high throughput hydroponics screening method to study the interaction between *P. medicaginis* and chickpea, which can be used for the identification of new PRR-resistant genotypes. The method can also be used for early generation screening of large breeding populations. We demonstrated the efficiency, reliability, and relevance of our hydroponics method by comparing the results of diverse genotypes and a RIL mapping population with field-based screening through both qualitative and quantitative measurements. Different phenotypic parameters used to select for PRR resistance showed a high correlation. This method will facilitate reliable and rapid screening of large germplasm collections and the development of new PRR-resistant cultivars.

## Figures and Tables

**Figure 1 plants-12-04069-f001:**
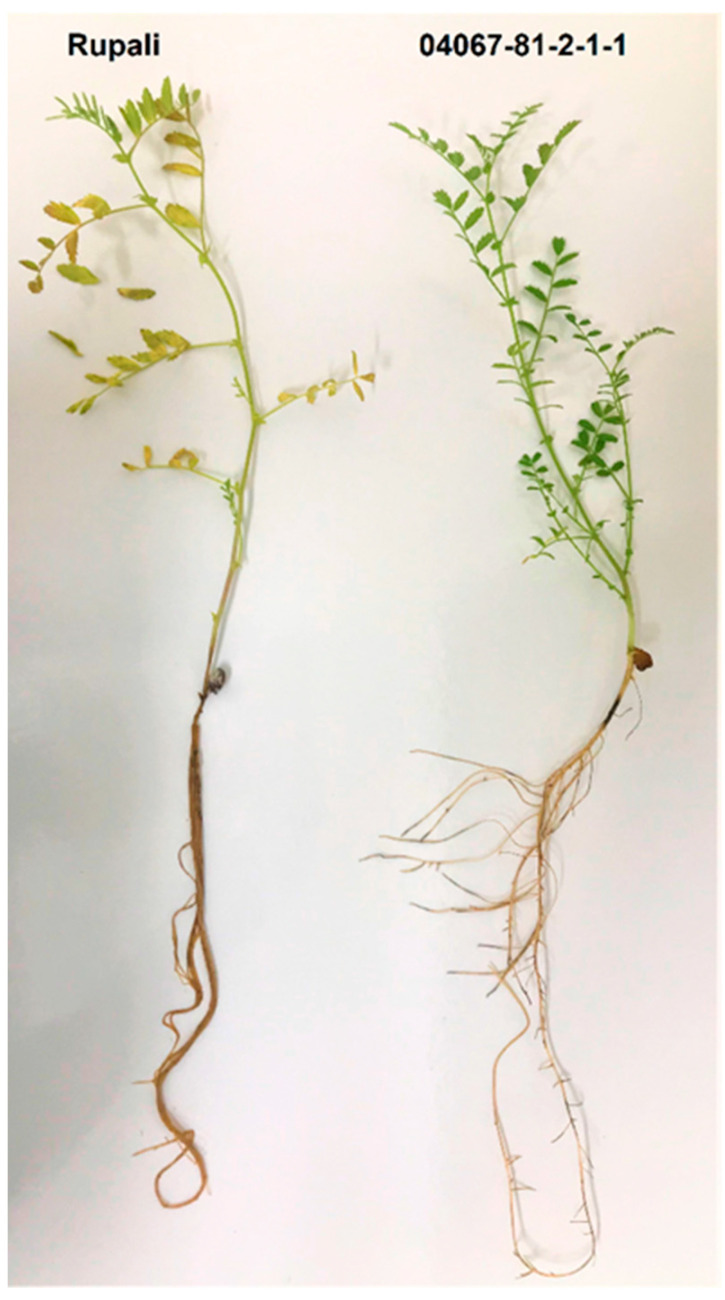
Phenotypic response of genotypes Rupali (**left**) and 04067-81-2-1-1 (**right**) 12 days after inoculation with *Phytophthora medicaginis* in E1.

**Figure 2 plants-12-04069-f002:**
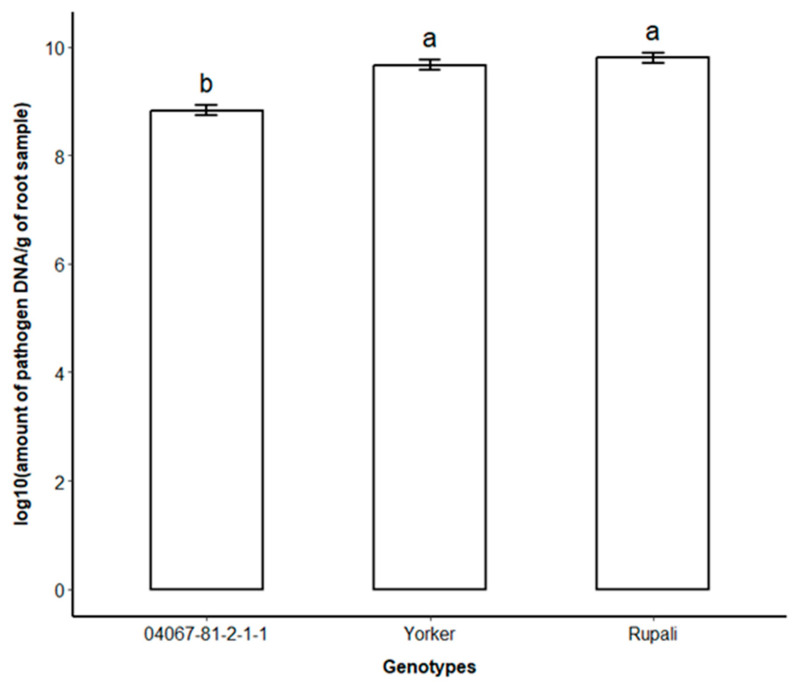
*Phytophthora medicaginis* DNA concentration (log transformed copies/g root) in roots of three chickpea genotypes 12 days after inoculation in hydroponics Experiment 1. Different letters are significantly different (*p* ≤ 0.05). Error bars represent the standard error of the mean of six replicates.

**Figure 3 plants-12-04069-f003:**
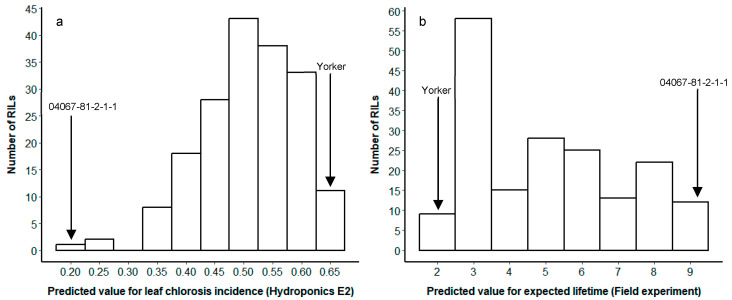
Frequency distributions of (**a**) predicted values of leaf-chlorosis incidence from hydroponic Experiment 2 and (**b**) expected lifetime from the field experiment in the Yorker × 04067-81-2-1-1 RIL population. Arrows’ positions indicate the parameter means for parents, Yorker, and 04067-81-2-1-1.

**Table 1 plants-12-04069-t001:** Penalized Quasi Likelihood estimates of the variance component associated with each random term expressed as a percentage of total variance, excluding the baseline variance for a logistic distribution of π^2^β.

Term	Leaf Chlorosis Incidence	Canker Incidence	Root Lesion Score
Rack	0	2.68	2.17
Rack:Column	7.16	3.53	0
Rack:Row	10.97	0	10.67
Genotype	79.75	93.88	87.16
Genotype:Rack	2.12	0	0

**Table 2 plants-12-04069-t002:** Published and unpublished PRR disease ratings and predicted genotypic effects (on the logit scale) for each of the three PRR-related parameters of 11 chickpea genotypes 12 days after inoculation with *P. medicaginis* in E1. Predicted genotypic effects (on the logit scale) and back-transformed values on the probability scale in brackets are presented. Note that for leaf-chlorosis incidence and canker incidence, these back-transformed probabilities are predicted probabilities of the relevant incidence. For the root lesions score, these are probabilities of a score of 1 or 2, i.e., the predicted probability of not exceeding a 50% root-lesion threshold. The r^2^ value is the mean (model-based) reliability of each parameter.

Genotype	Disease Rating	Reference	Leaf Chlorosis Incidence	Canker Incidence	Root Lesion Score, 1–4
04067-81-2-1-1	MR-MS	Hobson and Bithell, unpublished results [24]	−1.6 (0.26)	−2.44 (0.07)	5.34 (0.95)
CICA1328	MS	Hobson and Bithell, unpublished results [24]	−0.86 (0.42)	1.01 (0.72)	2.05 (0.42)
CICA1815	MS	Hobson and Bithell, unpublished results [24]	−0.25 (0.57)	−0.86 (0.28)	1.65 (0.32)
CBA Captain	S	NVT [25]	−0.04 (0.62)	0.32 (0.56)	−0.12 (0.08)
PBA HatTrick	S	NVT [25]	0.63 (0.76)	1.54 (0.81)	−1.54 (0.02)
PBA Seamer	S	NVT [25]	0.29 (0.69)	1.02 (0.72)	−0.24 (0.07)
Yorker	S	Hobson and Bithell, unpublished results [24]	0.41 (0.72)	0.74 (0.66)	−0.97 (0.03)
Kyabra	S-VS	NVT [25]	0.11 (0.66)	−0.03 (0.47)	−1.22 (0.03)
PBA Boundary	VS	NVT [25]	0.33 (0.70)	1.97 (0.87)	−1.34 (0.02)
PBA Monarch	VS	NVT [25]	0.17 (0.67)	0.87 (0.69)	−0.5 (0.05)
Rupali	VS	Hobson and Bithell, unpublished results [24]	0.81 (0.79)	1.53 (0.81)	−3.12 (0.00)
r^2^			0.88	0.84	0.85

VS = very susceptible, S = susceptible, MS = moderately susceptible, MR = moderately resistant.

**Table 3 plants-12-04069-t003:** Spearman rank correlation among the three parameters.

Term	Leaf Chlorosis Incidence	Canker Incidence	Root Lesion Score
Leaf chlorosis incidence	1		
Canker incidence	0.78 **	1	
Root lesion score	−0.97 ***	−0.82 **	1

** Significant at *p* = 0.01.; *** Significant at *p* = 0.001.

**Table 4 plants-12-04069-t004:** Penalized Quasi Likelihood estimates of variance parameters for additive and non-additive genetic terms in the combined analysis of hydroponics (Experiment 2) and the field experiment.

	Variance Parameter	
Experiment	Additive	Non-Additive
Field	1.15	2.8
Hydroponics	0.02	0.14
Correlation	−1	−0.35

## Data Availability

All data were reported in this paper and will also available if requested.

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
