# Peer review of "Rapid and High Throughput Hydroponics Phenotyping Method for Evaluating Chickpea Resistance to Phytophthora Root Rot"

_plants, 2023, doi:10.3390/plants12234069_

Round 1
Reviewer 1 Report
Comments and Suggestions for Authors
In this manuscript (plants-2722115) entitled "Rapid and high throughput hydroponics phenotyping method for evaluating chickpea resistance to Phytophthora root rot" submitted to Plants, Muhammad Ahsan Asif and colleagues have developed and evaluated a new space saving (400 plants/m2), rapid (<12 days) and simplified hydroponics based Phytophthora root rot (PRR) phenotyping method. This research is interesting and convincing, but minor points need to be addressed to improve the quality of this manuscript.
1. To my knowledge, plant resistance is affected by humidity. Authors should compare the PRR resistance between hydroponic plants and potting plants in the revised manuscript.
2. For Figures 1, at least three representative plants should be shown for phenotypic response of genotypes Rupali and 04067-81-2-1-1 to Phytophthora medicaginis inoculation in the revised Figure 1. In addition, microscopic analysis should be performed to show the P. medicaginis development.
3. For Figures 2, authors only show the data collected from one timepoint (12 days after inoculation). To show the pathogen development, at least three different time points should be analyzed in the revision.
4, To better understand this study, a model to depict the hydroponics based PRR phenotyping method should be included as a summary Figure in the revised manuscript.
Author Response
First, we would like to thank the reviewer for their valuable comments, which really helped us to improve the quality of this manuscript. We have made all the necessary changes in revised version and addressed all the comments in attached document. Please note that our responses to the reviewer comments are shown in bold text and our response in plain text.

Reviewer 2 Report
Comments and Suggestions for Authors
The manuscript titled "Rapid and high throughput hydroponics phenotyping method for evaluating chickpea resistance to Phytophthora root rot" by Asif et al. presents an interesting hydroponic Phytophthora root rot phenotyping method that bears several attractive characteristics such as being rapid and space saving. It also eliminates seedling transplant requirements following germination and preparation of a zoospore inoculum. The topic is certainly of interest to the readers of Plants and merits consideration for publication.
Abstract:
- written very well, it conveys the necessary amount of information without being overly verbose
Introduction:
- informative and concise, well referenced. The authors demonstrate their expertise in the field adequately.
Results:
- Please give a short introduction for each of the experiments so that the reader better understands the objectives. For example, 2.1 immediately begins with the reporting of the data; some context would be appreciated, especially to readers who are interested but not experts in the field.
- Fig. 3 seems quite small and of low resolution to me. Also, the additional box below the figure should be omitted in favor of the legend. Furthermore, descriptions and arrows are partially overlapping. Presentation of this figure needs to be improved.
Discussion:
- no criticisms or suggestions on my behalf
Materials and Methods:
- all information for the reproduction of the experiments is given
Supplementary information:
- The additional photographs are very helpful to describe the system the authors designed and tested
Recommendation:
- very good work, acceptable in the journal Plants after minor revision
Author Response

(The authors gave the same response as above.)

Reviewer 3 Report
Comments and Suggestions for Authors
I have two questions:
Were the experiments (E1, E2) carried out twice?
Has the use of zoospores been tested? Zoospores are the main infection units.

Author Response

(The authors gave the same response as above.)
